# GRAPHPROP: TRAINING THE GRAPH FOUNDATION MODELS USING GRAPH PROPERTIES

## ABSTRACT

In this work, we focus on training Graph Foundation Models (GFMs) for graph-level tasks like protein classification. Effective GFM training requires capturing information consistent across different domains. We have discovered that graph structures provide more consistent cross-domain information compared to node features and graph labels. However, traditional in-context learning methods primarily focus on transferring node features from various domains into a unified representation space but often lack structural cross-domain generalization. To address this, we introduce a method called GraphProp, which emphasizes structural generalization. The GraphProp training process consists of two main phases: initially, it trains a structural GFM through the supervised prediction of graph structural properties. It then uses the structural representation from this GFM as positional encoding to train a comprehensive GFM. This phase of training utilizes in-context learning with domain-specific node features and graph labels to improve cross-domain node feature generalization. Additionally, employing data augmentation in training the structural GFM helps address the scarcity of labeled graph data and facilitates explicit cross-domain structural generalization. Our experimental results demonstrate that GraphProp significantly outperforms traditional in-context learning methods, especially in handling graphs without node features.

## 1 INTRODUCTION

Graph Foundation Models (GFMs) are gaining more and more attention in research related to graph data, as they aim to leverage diverse data to improve effectiveness across various tasks and domains. Recent advancements (Galkin et al., 2023; Zheng et al., 2023) demonstrate that GFMs can generalize well to new, unseen graphs. These models are typically classified into three types based on their adaptability: domain-specific, task-specific, and primitive (Mao et al.). Domain-specific GFMs are designed to learn universal features within a particular domain, allowing a single model to efficiently handle multiple tasks, often outperforming specialized models. Examples include Mole-BERT (Xia et al., 2023), DPA-2 (Zhang et al., 2023), and DiG (Zheng et al., 2023), which are developed to handle multiple molecular tasks within the chemical domain. Task-specific GFMs are trained on rich data sources to perform specific tasks, making them suitable for fields with less abundant data. For example, ULTRAQUERY (Galkin et al., 2024) and ULTRA (Galkin et al., 2023) are designed for knowledge graph reasoning and can be trained on extensive data sources like Wikipedia, enabling their application in less data-rich domains. Similarly, GraphFM (Lachi et al., 2024), GraphAny (Zhao et al., 2024), and GRAPHTEXT (Zhao et al., 2023b) are trained for node classification tasks using graphs from various domains, including chemical and social networks. Primitive GFMs are more versatile but limited in the range of datasets and tasks they can handle. For instance, UniAug (Tang et al., 2024b), a universal graph structure augmentor based on a diffusion model, captures diverse graph data patterns and can be used to adaptively assist downstream tasks.

The main challenge in developing GFMs is capturing consistent information from graph data that varies across different domains (Galkin et al., 2023). For instance, in molecular data (Yang et al., 2016), graph structures represent 3-D spatial relationships and atomic bonds, while node features capture chemical properties. In social networks (Dwivedi et al., 2023), structures represent user connections, and node features reflect user attributes. Due to these differing distributions, it is challenging for a single model to learn unified representations across domains. Traditionally, one approach is to convert graph structures and node features into text and leverage large language mod-

els (LLMs) to create unified representations for graphs from different domains. Although no direct method currently exists for learning unified graph structure representations, existing graph reasoning LLMs can be adapted for this purpose. For example, GraphQA (Fatemi et al., 2023) describes graph connectivity in text and then poses graph reasoning questions to LLMs. By incorporating domain descriptions into these prompts, GraphQA can be adapted to learn unified representations of graph structures, effectively functioning as a structural GFM. Similarly, other graph reasoning LLMs, like NLGraph (Wang et al., 2024), which focuses on tasks such as shortest paths and connectivity by converting graphs into text, can also be used to learn unified graph structure representations. For learning unified node feature representations, the One For All (OFA) method proposed by (Liu et al., 2023a) employs text-attributed graphs (TAGs) to consolidate graph datasets from various domains into a single, large TAG dataset, and then utilizes LLMs to learn unified node feature representations across all domains jointly. However, these methods have limitations. Graph reasoning GFMs primarily focus on reasoning abilities rather than comprehensive structural representations, and in-context GFMs may struggle with structural generalization, especially when dealing with graph data lacking node features.

To improve GFMs, we aim to capture information that remains consistent across different domains. We have observed that the structure of graphs contains invariant information (properties depending on the abstract structure only) that is shared across domains. For example, whether dealing with molecular data or social networks, their abstract graph structures exhibit common properties like the fractional chromatic number (Scheinerman & Ullman, 2013) and Lovász number (Lovász, 1979), even if their specific values differ. On the other hand, node features and graph labels are highly domain-specific and lack this cross-domain consistency. For instance, node features in molecular data describe chemical properties, while in social networks, they represent user attributes, without overlap between them. Similarly, graph labels, such as the class of a molecule or the type of social community, are tied to domain-specific knowledge, making them unique to each domain.

Given these distinctions, we introduce GraphProp, a GFM training method that separates the use of structural information from domain-specific node features and graph labels. GraphProp begins by training a cross-domain structural GFM through the prediction of graph properties in a supervised manner. To achieve comprehensive cross-domain structural representations, GraphProp incorporates a wide range of graph properties, including novel ones like the fractional chromatic number. This approach enables explicit unified graph structure learning through graph data augmentation, extending traditional methods like G-Mixup (Han et al., 2022) for cross-domain GFM training. GraphProp also addresses the scarcity of graph data by leveraging unlabeled and synthetic graphs, ensuring sufficient data for effective GFM training. After training the structural GFM, we use its structural representation as positional encoding to train a comprehensive GFM. This training phase employs in-context learning with domain-specific node features and graph labels to enhance cross-domain node feature generalization. Overall, the GraphProp framework achieves both structural and node feature generalization across domains.

Our contributions are as follows:

- We introduce GraphProp, a GFM training method that first trains a structural GFM by predicting graph properties. Then, it uses these structural representations to train a comprehensive GFM through in-context learning with domain-specific features and labels.

- To the best of our knowledge, GraphProp is the first GFM that achieves both structural and node feature generalization across domains for graph-level tasks. Many existing GFMs merely use in-context learning with text-attributed graphs, lacking structural generalization.

- We bridge the use of graph theory in GFM training through graph property prediction. This approach addresses the scarcity of labeled data by effectively utilizing unlabeled and even synthetic graphs for scalable GFM training.

## 2 PRELIMINARY

### 2.1 NOTATIONS

In this work, we use $x$, $\mathbf{x}$, $\mathbf{X}$, and $\mathcal{X}$ to denote a scalar, vector, matrix, and set, respectively. We define $[n] = \{1, 2, \ldots, n\}$. Let $G = (V, E)$ be a graph with $n$ nodes and node features $\{\mathbf{x}_v \in$

$\mathbb{R}^d \mid v \in V\}$. We denote the adjacency matrix as $\mathbf{A} \in \{0,1\}^{n \times n}$, the node features matrix as $\mathbf{X} = [\mathbf{x}_1, \ldots, \mathbf{x}_n]^\top \in \mathbb{R}^{n \times d}$, and the graph label as $y$. Let $\mathbb{G} := \{\mathcal{G}^{(1)}, \ldots, \mathcal{G}^{(M)}\}$ be the graph datasets from $M$ different domains and denote $G^{(m)} \in \mathcal{G}^{(m)}$ a graph from the $m$-th domain, with adjacency matrix $\mathbf{A}^{(m)}$, node feature matrix $\mathbf{X}^{(m)}$, and graph label $y^{(m)}$. For simplicity and without loss of generality, we assume all graphs have the same number of nodes $n$, all domains have $N$ graphs, and all domains have the same feature dimension $d$. The actual values will be specified in our experiments.

## 2.2 GRAPH PROPERTIES

In graph theory, a graph property, or invariant, depends solely on the structure of graph, not on its representation or labeling. Given a graph $G$, a function $\alpha : \mathbb{A} \to \mathbb{R}$ maps the adjacency matrix $\mathbf{A}$ to a real number, representing a specific graph property $p$. In our study, we consider $K$ different graph properties to form a property vector $\mathbf{p}$, where each property $p_k$ is computed by a known algorithm $\alpha_k(\mathbf{A})$. This can be expressed as:

$$\mathbf{p} = \boldsymbol{\alpha}(\mathbf{A}), \quad \text{where} \quad \mathbf{p} = [p_1, \ldots, p_K]^\top \quad \text{and} \quad p_k = \alpha_k(\mathbf{A}). \tag{1}$$

Because the scales of graph properties can vary, we normalize each property $p_k$ to ensure each is equally important in the vector $\mathbf{p}$. With a dataset of $N$ graphs, the $k$-th property of the $i$-th graph is $p_k^i$. We normalize $p_k$ as follows:

$$p_k = \frac{p_k^i - \bar{p}_k}{\sigma_k}, \quad \text{where} \quad \bar{p}_k = \frac{1}{N} \sum_{i=1}^N p_k^i, \quad \text{and} \quad \sigma_k = \sqrt{\frac{1}{N} \sum_{i=1}^N (p_k^i - \bar{p}_k)^2} \tag{2}$$

Some graph properties require NP-hard algorithms to compute, such as the independent number Biggs (1993) and clique number (Aigner, 1995) in Table 5 and Table 6. Others can be computed using polynomial-time algorithms, like the fractional chromatic number (Scheinerman & Ullman, 2013) and Lovász number (Lovász, 1979) in Table 4. We will provide detailed descriptions of these properties, particularly those computable in polynomial time, in the Appendix B.1.

## 2.3 TRANSFORMER

The transformer model (Vaswani, 2017) consists of a self-attention mechanism and a feed-forward network (FFN). Let $\mathbf{H} = [\mathbf{h}_1, \ldots, \mathbf{h}_n]^\top \in \mathbb{R}^{n \times d}$ represent the matrix of hidden states, where each $\mathbf{h}_i$ is the hidden state at position $i$. This matrix is projected into three matrices: queries $\mathbf{Q}$, keys $\mathbf{K}$, and values $\mathbf{V}$, using the projection matrices $\mathbf{W}_Q \in \mathbb{R}^{d \times d_Q}$, $\mathbf{W}_K \in \mathbb{R}^{d \times d_K}$, and $\mathbf{W}_V \in \mathbb{R}^{d \times d_V}$, respectively, i.e., $\mathbf{Q} = \mathbf{H}\mathbf{W}_Q$, $\mathbf{K} = \mathbf{H}\mathbf{W}_K$, $\mathbf{V} = \mathbf{H}\mathbf{W}_V$. The self-attention mechanism is then computed as:

$$\text{attn}(\mathbf{H}) = \text{softmax}\left(\frac{\mathbf{Q}\mathbf{K}^\top}{\sqrt{d_K}}\right) \mathbf{V}. \tag{3}$$

The transformer updates the input through the following function:

$$T(\mathbf{H}) = \text{Norm}(\hat{\mathbf{H}} + \text{FFN}(\hat{\mathbf{H}})), \quad \text{where} \quad \hat{\mathbf{H}} = \text{Norm}(\mathbf{H} + \text{attn}(\mathbf{H})). \tag{4}$$

## 2.4 GRAPH TRANSFORMER

Graph transformers extend traditional transformers by integrating graph structural information through positional encodings (Black et al., 2024). Let $\mathbf{B} = [\mathbf{b}_1, \ldots, \mathbf{b}_n]^\top \in \mathbb{R}^{n \times d}$ represent the positional encoding matrix, where each $\mathbf{b}_i \in \mathbb{R}^d$ is the positional encoding for node $i$. The function $\phi : \mathbb{R}^{n \times n} \to \mathbb{R}^{n \times d}$ computes $\mathbf{B} = \phi(\mathbf{A})$. To enhance node features, the positional encodings can be either concatenated with or added to the original features, i.e.,

$$\text{Concatenate:} \quad \hat{\mathbf{x}}_i = \mathbf{x}_i \oplus \mathbf{b}_i, \quad \text{or} \quad \text{Add:} \quad \hat{\mathbf{x}}_i = \mathbf{x}_i + \mathbf{b}_i, \quad \forall\, i \in [n] \tag{5}$$

The graph transformer then uses the augmented feature matrix $\hat{\mathbf{X}} = [\hat{\mathbf{x}}_1, \ldots, \hat{\mathbf{x}}_n]^\top$ as the input of the transformer $T(\cdot)$. A variant using relative positional encodings is discussed in the Appendix B.5.

**Definition 2.1** (Invertible Positional Encoding). A positional encoding matrix $\mathbf{B}$ is invertible if there exists a mapping $\phi^{-1}$ such that $\mathbf{A} = \phi^{-1}(\mathbf{B})$.

## 2.5 IN-CONTEXT LEARNING

In-context learning is a widely used method for creating unified graph representations across different domains. It works by describing graph structures or node features in text, incorporating domain-specific details, and using a LLM to generate these unified representations. This technique has been applied to both graph structures and node features.

**Text-Structure Graphs (TSG)**  Building on GraphQA Fatemi et al. (2023), we introduce TSG to represent the adjacency matrix $\mathbf{A}$ of graph $G$ using descriptive text prompts. For example:

> **Type**: TSG; **Domain**: Molecular; **Number of Nodes**: 10; **Overall Properties**: Fiedler value = 0.85; Lovász number = 1.67; **Connectivity**:
> - Node 1: Connected to nodes 2 and 3;
> - Node 2: Connected to nodes 6 and 8; . . .

**Text-Attributed Graphs (TAG)**  Building on OFA (Liu et al., 2023a), we introduce TAG to represent the node feature matrix $\mathbf{X}$ of graph $G$ using descriptive text prompts. For example:

> **Type**: TAG; **Domain**: Molecular; **Number of Nodes**: 10; **Overall Chemical Features**: Polar, Aromatic, Hydrophobic regions; **Node Features**:
> - Node 1: Atom: Carbon, sp3 hybridization, helix chirality, . . . ;
> - Node 2: Atom: Oxygen, involved in a hydrogen bond, . . . ; . . .

**Learning Unified Representations**  Consider a graph $G^{(m)}$ from domain $m$. We define its unified graph structure representation as $\mathbf{c}^{(m)}$ and its unified node feature representation matrix as $\mathbf{E}^{(m)}$. These unified representations are generated by a LLM as follows:

$$\mathbf{c}^{(m)} = \text{LLM}(\text{TSG of } G^{(m)}) \quad \text{and} \quad \mathbf{E}^{(m)} = \text{LLM}(\text{TAG of } G^{(m)}). \tag{6}$$

Since $\mathbf{E}^{(m)} = [\mathbf{e}_1^{(m)}, \ldots, \mathbf{e}_n^{(m)}]^\top$ represents the node-level features, we can compute the graph-level representation as the average of all node features: $\bar{\mathbf{e}}^{(m)} = \frac{1}{n} \sum_{i=1}^n \mathbf{e}_i^{(m)}$. Research in in-context learning (Fatemi et al., 2023; Liu et al., 2023a) suggests that these unified representations, even from different domains, share common representation spaces.

## 3 GRAPHPROP METHODS

In this section, we present the GraphProp method, covering its motivation, the structural GFM training and the comprehensive GFM training.

### 3.1 MOTIVATIONS

Our goal is to train GFMs that effectively learn across different domains by capturing consistent information shared among them. This raises the question: **How much cross-domain consistent information do graph structures and node features contain, respectively?** We intuitively believe that graph structures hold more cross-domain consistent information than node features. For example, both molecular data and social networks share common graph properties like the Lovász number, even if the specific values differ. In contrast, node features are highly domain-specific—molecular data features describe chemical properties, while social network features relate to user attributes, with little overlap between them.

To quantify this, we used in-context learning (Section 2.5) to obtain graph structure representations $\mathbf{c}_i^{(m)}$ and average node feature representations $\bar{\mathbf{e}}_i^{(m)}$ for each graph $G_i^{(m)}$. In-context learning ensures that all graph structure representations are sampled from the same distribution, $\mathbf{c}_i^{(m)} \sim \mathcal{D}_c$, and similarly, $\bar{\mathbf{e}}_i^{(m)} \sim \mathcal{D}_e$. We normalized these distributions to have zero means, i.e., $\mathbb{E}(\mathbf{c}_i^{(m)}) = 0$ and

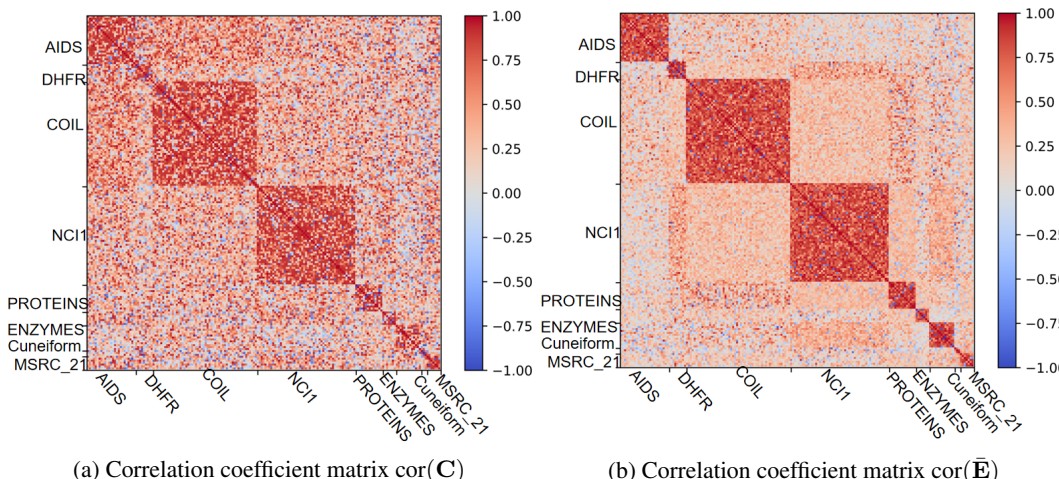

(a) Correlation coefficient matrix $\mathrm{cor}(\mathbf{C})$      (b) Correlation coefficient matrix $\mathrm{cor}(\bar{\mathbf{E}})$

Figure 1: In-domain and cross-domain correlation coefficients of the representations given by in-context learning on eight graph datasets.

$\mathbb{E}(\bar{\mathbf{e}}_i^{(m)}) = 0$. The correlation coefficient between any two representations is defined as:

$$\rho(\mathbf{c}_i^{(m_1)}, \mathbf{c}_j^{(m_2)}) = \frac{\langle \mathbf{c}_i^{(m_1)}, \mathbf{c}_j^{(m_2)} \rangle}{\|\mathbf{c}_i^{(m_1)}\| \|\mathbf{c}_j^{(m_2)}\|} \quad \text{and} \quad \rho(\bar{\mathbf{e}}_i^{(m_1)}, \bar{\mathbf{e}}_j^{(m_2)}) = \frac{\langle \bar{\mathbf{e}}_i^{(m_1)}, \bar{\mathbf{e}}_j^{(m_2)} \rangle}{\|\bar{\mathbf{e}}_i^{(m_1)}\| \|\bar{\mathbf{e}}_j^{(m_2)}\|}. \tag{7}$$

When $m_1 = m_2$, these measure in-domain correlations; when $m_1 \neq m_2$, these measure cross-domain correlations. We compiled the representations into matrices $\mathbf{C} = [\mathbf{c}_1^{(1)}, \ldots, \mathbf{c}_N^{(M)}]^\top \in \mathbb{R}^{MN \times d}$ and $\bar{\mathbf{E}} = [\bar{\mathbf{e}}_1^{(1)}, \ldots, \bar{\mathbf{e}}_N^{(M)}]^\top \in \mathbb{R}^{MN \times d}$. The correlation coefficient matrices $\mathrm{cor}(\mathbf{C}) \in [-1, 1]^{MN \times MN}$ and $\mathrm{cor}(\bar{\mathbf{E}}) \in [-1, 1]^{MN \times MN}$ capture the pairwise correlations of rows in $\mathbf{C}$ and $\bar{\mathbf{E}}$. Visualizations given by Figure 1 show in-domain correlations on the diagonal blocks and cross-domain correlations on the off-diagonal blocks. We observed that the cross-domain correlation of $\mathbf{C}$ is higher than that of $\bar{\mathbf{E}}$, indicating that graph structures contain more cross-domain consistent information than node features. Furthermore, the low cross-domain correlation of $\bar{\mathbf{E}}$ suggests that node features have little cross-domain consistent information. These results confirm our intuition and match real-world observations, reinforcing the importance of focusing on graph structures in GFM training.

To address this, we first trains a structural GFM by predicting graph properties. Then, we uses these structural representations to train a comprehensive GFM through in-context learning with domain-specific features and labels. The detailed steps are outlined as follows.

## 3.2 TRAINING A STRUCTURAL GFM

**Training** We begin by calculating a ground truth graph properties vector $\mathbf{p}$ using established graph theory algorithms (see Section 2.2). The goal is to train a structural GFM to predict this vector. Let $\mathbf{B} \in \mathbb{R}^{n \times d}$ be the positional encoding matrix, computed as $\mathbf{B} = \phi(\mathbf{A})$. The structural GFM, denoted as $f(\cdot; \Theta)$, is implemented using graph transformers with parameters $\Theta$. Since node features $\mathbf{X}$ are not used during training, we feed the positional encoding matrix $\mathbf{B}$ directly into $f(\cdot; \Theta)$, which generates a structural representation $\mathbf{Z} \in \mathbb{R}^{n \times d}$. The graph properties are then predicted using a regressor $\varphi(\cdot; \Psi)$, with parameters $\Psi$:

$$\hat{\mathbf{p}}_{\Theta, \Psi} = \varphi(\mathbf{Z}_\Theta; \Psi), \quad \text{where} \quad \mathbf{Z}_\Theta = f(\mathbf{B}; \Theta). \tag{8}$$

We denote $\ell_{\text{prop}}(\cdot, \cdot)$ the graph property regression loss. During the training process, we optimize the parameters $\Theta$ and $\Psi$ by solving the minimization problem presented below:

$$\Theta^*, \Psi^* = \underset{\Theta, \Psi}{\arg\min} \, \ell_{\text{prop}}(\hat{\mathbf{p}}_{\Theta, \Psi}, \mathbf{p}) \tag{9}$$

where $\ell_{\text{prop}}(\hat{\mathbf{p}}_{\Theta, \Psi}, \mathbf{p}) = \|\hat{\mathbf{p}}_{\Theta, \Psi} - \mathbf{p}\|^2$ for example.

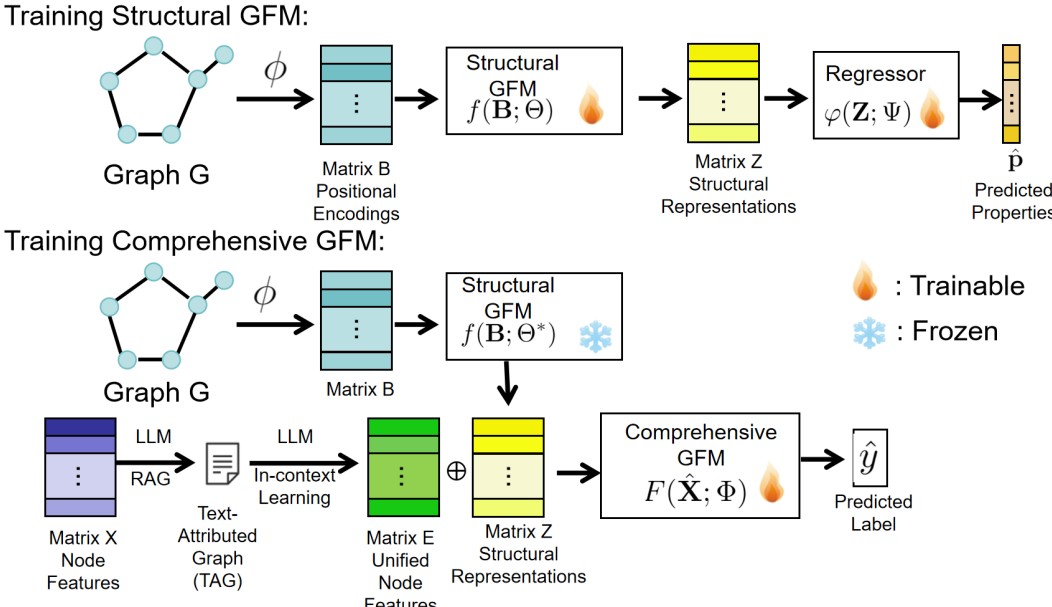

Figure 2: Framework of GraphProp

It is essential that the positional encoding matrix $\mathbf{B}$ used for graph property prediction must be invertible (see Definition 2.1). Invertibility ensures that graph properties can be accurately predicted as $\mathbf{p} = \boldsymbol{\alpha}(\phi^{-1}(\mathbf{B}))$. If $\mathbf{B}$ is not invertible, it may fail to capture all the necessary information from $\mathbf{A}$, leading to inaccurate predictions. For example, spectral embedding is a common positional encoding in graph transformers. Let $\mathbf{D} = \mathrm{diag}(\sum_j \mathbf{A}_{ij})$ be the degree matrix and $\mathbf{L} = \mathbf{D} - \mathbf{A}$ the Laplacian matrix, with singular value decomposition $\mathbf{L} = \mathbf{U}\boldsymbol{\Lambda}\mathbf{U}^\top$. The eigenvectors $\mathbf{U}$ corresponding to the top-k eigenvalues are often used as the positional encodings. However, this specrtal embedding is not invertible because $\mathbf{A}$ cannot be reconstructed from $\mathbf{U}$. Therefore, spectral embedding is not effective for property prediction. In contrast, the positional encoding $\mathbf{B} = \mathbf{U}\boldsymbol{\Lambda}^{1/2}$ is invertible and can be used for effective property prediction.

**Data Augmentation** Given two graphs $G^{(m_1)}$ and $G^{(m_2)}$ with adjacency matrices $\mathbf{A}^{(m_1)}$ and $\mathbf{A}^{(m_2)}$, we can create a cross-domain augmented graph $\hat{G}$ with an adjacency matrix $\hat{\mathbf{A}}$ using a mixup technique: $\hat{\mathbf{A}} = \mathrm{mixup}(\mathbf{A}^{(m_1)}, \mathbf{A}^{(m_2)})$ (see Section B.4). We then compute its graph properties and incorporate them into the GFM training process, enhancing the model's ability to learn cross-domain invariant structural information within our GraphProp framework. Traditional data augmentation methods, like G-Mixup (Han et al., 2022), are effective when the graphs come from the same domain, requiring the creation of a soft label $\hat{y}$ (as in equation 22) for training. However, this approach fails when combining graphs from different domains, as mixing labels from unrelated domains doesn't make sense.

To extend structural GFM training to handle graphs from unseen domains beyond the $M$ domains in the training dataset, we can randomly generate adjacency matrices and use them for property prediction. Previous methods couldn't utilize synthetic graphs due to the absence of labels and task context. However, our property prediction approach allows us to include these synthetic graphs in the training process, further strengthening the structural learning capabilities of GFMs.

### 3.3 TRAINING A COMPREHENSIVE GFM

In this section, we train a comprehensive GFM using in-context learning (see Section 2.5). Given the trained structural GFM $f$ with parameters $\Theta^*$, we compute the positional encoding $\mathbf{B}$ for each graph $G$ to obtain the structural representation $\mathbf{Z}$:

$$\mathbf{Z} = f(\mathbf{B}; \Theta^*) \quad \text{where} \quad \mathbf{B} = \phi(\mathbf{A}). \tag{10}$$

Let $\mathbf{E} = [\mathbf{e}_1, \ldots, \mathbf{e}_n]^\top$ be the unified node features obtained from equation 6. We can create an augmented feature matrix $\hat{\mathbf{X}}$ by combining the unified node features $\mathbf{e}_i$ with the corresponding structural representation $\mathbf{z}_i$:

$$\hat{\mathbf{x}}_i = \mathbf{e}_i \oplus \mathbf{z}_i, \quad \forall i \in [n]. \tag{11}$$

Next, we train the comprehensive GFM $F(\cdot; \Phi)$ with trainable parameters $\Phi$ by minimizing the cross-entropy loss $\ell_{\text{ce}}$ for classification:

$$\Phi^* = \underset{\Phi}{\arg\min} \, \ell_{\text{ce}}(\hat{y}_\Phi, y), \quad \text{where} \quad \hat{y}_\Phi = F(\hat{\mathbf{X}}; \Phi). \tag{12}$$

## 3.4 ADVANTAGES

- **Structural and Node Feature Cross-Domain Generalization:** To the best of our knowledge, GraphProp is the first GFM that achieves both structural and node feature generalization across domains, designed for graph-level tasks. It learns unified structural representations $\mathbf{Z}$ through property prediction and unified node features via in-context learning on TAG, enabling it to handle various graph types, including those without node features.

  Many existing GFMs prioritize node feature generalization through in-context learning but often overlook structural generalization. For instance, OFA (Liu et al., 2023a) performs well with node features but struggles with graphs that lack them. Models like GraphQA Fatemi et al. (2023) try to achieve structural generalization by reasoning with TSGs and posing simple questions to LLMs, yet they mainly focus on enhancing reasoning rather than developing comprehensive structural representations. Converting complex graph structures into text can result in the loss of essential information about the graph's overall properties. In contrast, our structural GFM directly regresses graph properties without relying on TSGs, enabling effective cross-domain structural generalization, as these properties are topological characteristics present across various domains.

- **Bridging GFMs and Graph Theory:** GraphProp is a self-learning method that leverages a wide array of graph properties from graph theory, enabling the structural GFM to learn comprehensive representations $\mathbf{Z}$. Beyond graph-level properties, this method can also predict node-level properties (e.g., degree and centrality in Table 8) and node-pair properties (e.g., shortest path (Schrijver, 2012)). More details are available in Appendix B.1.

- **Addressing Data Scarcity:** Training foundation models usually requires large amounts of labeled data, which can be hard to find. In contrast, unlabeled data is much more plentiful. Similar to how LLMs like GPT (Radford, 2018) are pre-trained on unlabeled data by predicting the next word in a sequence, GraphProp uses unlabeled graph data for training structural GFMs through property prediction. Additionally, for large structural GFMs, data augmentation can create synthetic graphs to support scalable GFM training.

## 3.5 LIMITATIONS

- **Limited Scalability for Large Node-Level Tasks:** GraphProp is primarily designed for graph-level tasks and may struggle with large-scale node-level tasks, such as those involving graphs with many nodes (e.g., ogbn-arxiv (Wang et al., 2020)). Some graph property computations have polynomial complexity, while others are NP-hard, making them inefficient for very large graphs.

- **Graph Property Requirements:** Certain graph properties, like graph diameter, apply only to specific types of graphs, such as connected graphs. This means that graphs must be checked before certain properties can be used.

- **Addressing Domain-Specific Node Features:** While GraphProp can generate synthetic graphs to alleviate the scarcity of structural data, it does not address the lack of domain-specific node features, which requires specialized knowledge.

# 4 RELATED WORKS

Due to limited space, we have included graph properties, GFMs, graph transformers, graph reasoning methods, data augmentation and other graph theory benchmarks in the Appendix B.

## 5 EXPERIMENTS

In this section, we evaluate the cross-domain generalization of GraphProp in supervised learning and few-shot learning.

### 5.1 EXPERIMENT SETTINGS

**Graph Properties:** All the graph properties introduced in Appendix B.1 can be used in GraphProp. To simplify implementation, we selected fifteen properties with polynomial-time complexity, as listed in Table 4.

**Dataset:** We divided the dataset into two groups based on whether they have node features. The first group $\mathbb{G}_1$ includes datasets with node features: PROTEINS, NCI1, AIDS, HIV, and PCBA. The second group $\mathbb{G}_2$ includes datasets without node features: COLLAB, IMDB-B, DD, REDDIT-B, and REDDIT-M5K. HIV, and PCBA are from the OGB dataset Hu et al. (2020) and other from TUDataset Morris et al. (2020).

Table 1: Statistics of Datasets

| Name | # of graphs | # of classes | # of nodes | node attributes |
|------|------|------|------|------|
| PROTEINS | 1113 | 2 | 39.1 | yes |
| NCI1 | 4110 | 2 | 29.9 | yes |
| AIDS | 2000 | 2 | 15.69 | yes |
| HIV | 41127 | 2 | 25.5 | yes |
| PCBA | 437929 | 128 | 26.0 | yes |
| COLLAB | 5000 | 3 | 74.49 | no |
| IMDB-B | 1000 | 2 | 19.8 | no |
| DD | 1178 | 2 | 284.32 | no |
| REDDIT-B | 2000 | 2 | 429.63 | no |
| REDDIT-M5K | 4999 | 5 | 508.52 | no |

**Baselines:** Many GFMs for graph-level tasks are tailored to specific domains and are not suitable as baselines for cross-domain graph tasks. For instance, models like LLM4Mol (Qian et al., 2023) and GIMLET (Zhao et al., 2023a) are designed specifically for the molecular domain. Thus, we choose to use OFA with different LLMs as our baseline. Additionally, we included GNNs for comparison, such as a 5-layer GCN and a 3-layer Graph Transformer. Following OFA (Liu et al., 2023a), we selected three popular LLMs for both GraphProp and OFA: Sentence Transformer (st) (Reimers, 2019), e5-large-v2 (e5) (Wang et al., 2022), and Llama2-7b(Touvron et al., 2023).

**Structure:** Both the structural GFM $f(\cdot; \Theta)$ and the comprehensive GFM $F(\cdot; \Phi)$ are 3-layer graph transformers. The properties regressor $\varphi(\cdot; \Psi)$ is implemented with a 1-layer graph transformer followed by a 3-layer DNN. The structural representation $\mathbf{Z}$ has a dimension of 128, and the unified node representations $\mathbf{E}$ from the LLM in equation 6 are also 128, making the augmented feature $\hat{\mathbf{X}}$ have a total dimension of 256.

### 5.2 CROSS-DOMAIN SUPERVISED LEARNING

We ran experiments on supervised learning using all datasets from $\mathbb{G}_1$ and $\mathbb{G}_2$, training GFMs separately for each group. Each experiment used 10-fold cross-validation, with 80% for training, 10% for validation, and 10% for testing. The results are presented in Tables 2 and 3. To visualize overall performance, we plotted the average results for each group in subfigure (a) of Figure 3. In $\mathbb{G}_1$ (datasets with node features), GraphProp slightly outperforms OFA. In $\mathbb{G}_2$ (datasets without node features), GraphProp performs significantly better than OFA. This difference arises because OFA's in-context learning for a graph $G$ is defined as follows:

$$\hat{y} = \text{GNN}(\mathbf{A}, \mathbf{E}), \quad \text{where} \quad \mathbf{E} = \text{LLM}(\text{TAG of } G). \tag{13}$$

OFA's generalization depends on LLMs processing the TAG of $G$. In the case of $\mathbb{G}_2$, where node features are missing, there is no detailed TAG available. Instead, it only uses basic node features, such as degrees, which limits its generalization capabilities and reduces it to a basic GNN. In contrast, GraphProp's generalization benefits from both in-context learning and the structural GFM $f$. This enables GraphProp to capture cross-domain information from both node features and graph structure, making it applicable to graph datasets that lack node features.

### 5.3 CROSS-DOMAIN FEW-SHOT LEARNING

We conducted few-shot and zero-shot experiments using datasets from $\mathbb{G}_1$ and $\mathbb{G}_2$, training GFMs separately for each group. In the transfer scenario, both the test graphs and classes are unseen during

Table 2: Results of supervised learning on data group $\mathbb{G}_1$. The largest value is **bold**.

| Data | PROTEINS | NCI1 | AIDS | HIV | PCBA |
|------|----------|------|------|-----|------|
| Metric | ACC ↑ | ACC↑ | APR↑ | AUC↑ | APR↑ |
| GCN | $74.66 \pm 1.73$ | $75.81 \pm 1.28$ | $57.45\pm1.71$ | $74.21\pm1.55$ | $21.53\pm0.69$ |
| GT | $75.73 \pm 1.14$ | $76.39 \pm 1.52$ | $58.64\pm1.57$ | $75.86\pm1.09$ | $23.15\pm0.55$ |
| OFA-st | $78.61 \pm 2.35$ | $79.95 \pm 1.67$ | $61.91\pm1.96$ | $78.04\pm1.26$ | $21.86\pm0.73$ |
| OFA-e5 | $80.24 \pm 1.08$ | $81.77 \pm 0.92$ | $58.24\pm0.75$ | $76.22\pm1.93$ | $24.11\pm0.48$ |
| OFA-llama2 | $79.66 \pm 1.42$ | $80.07 \pm 1.35$ | $60.03\pm1.17$ | $77.52\pm1.88$ | $22.35\pm0.62$ |
| GraphProp-st | $\mathbf{83.12 \pm 1.89}$ | $\mathbf{84.79 \pm 1.27}$ | $61.64\pm1.27$ | $\mathbf{79.49\pm0.57}$ | $23.07\pm0.14$ |
| GraphProp-e5 | $82.63 \pm 1.25$ | $83.43 \pm 2.06$ | $\mathbf{64.07\pm0.93}$ | $78.17\pm1.34$ | $22.65\pm0.73$ |
| GraphProp-llama2 | $81.45 \pm 1.60$ | $81.15 \pm 1.30$ | $63.19\pm1.38$ | $78.54\pm1.42$ | $\mathbf{24.65\pm0.61}$ |

Table 3: Results of supervised learning on data group $\mathbb{G}_2$. The largest value is **bold**.

| Data | COLLAB | IMDB-B | DD | REDDIT-B | REDDIT-M5K |
|------|--------|--------|-----|----------|------------|
| Metric | ACC ↑ | ACC↑ | ACC↑ | ACC↑ | ACC↑ |
| GCN | $72.52 \pm 1.47$ | $76.39 \pm 1.47$ | $75.62 \pm 1.49$ | $77.45 \pm 1.31$ | $51.24 \pm 1.42$ |
| GT | $74.86 \pm 2.35$ | $77.13 \pm 1.32$ | $74.55 \pm 1.31$ | $78.52 \pm 1.14$ | $52.69 \pm 1.75$ |
| OFA-st | $74.24 \pm 1.43$ | $75.92 \pm 1.58$ | $77.34 \pm 1.14$ | $80.03 \pm 1.22$ | $53.28 \pm 1.24$ |
| OFA-e5 | $76.25 \pm 1.09$ | $78.19 \pm 1.33$ | $76.65 \pm 1.23$ | $79.62 \pm 1.17$ | $55.17 \pm 1.59$ |
| OFA-llama2 | $75.44 \pm 1.37$ | $77.69 \pm 1.25$ | $75.46 \pm 1.80$ | $78.23 \pm 1.35$ | $54.26 \pm 1.13$ |
| GraphProp-st | $79.27 \pm 1.14$ | $\mathbf{85.12 \pm 1.31}$ | $81.43 \pm 1.25$ | $82.69 \pm 1.52$ | $58.67 \pm 1.22$ |
| GraphProp-e5 | $81.35 \pm 1.32$ | $82.78 \pm 1.85$ | $\mathbf{82.31 \pm 1.41}$ | $\mathbf{85.32 \pm 1.17}$ | $59.36 \pm 1.27$ |
| GraphProp-llama2 | $\mathbf{82.64 \pm 1.58}$ | $83.42 \pm 1.70$ | $80.25 \pm 1.38$ | $84.38 \pm 1.26$ | $\mathbf{60.93 \pm 1.45}$ |

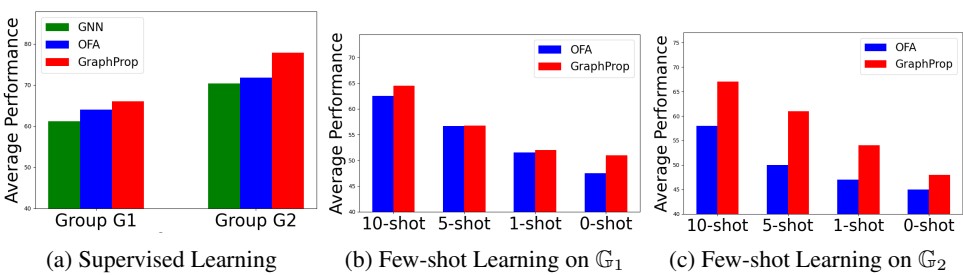

(a) Supervised Learning     (b) Few-shot Learning on $\mathbb{G}_1$     (c) Few-shot Learning on $\mathbb{G}_2$

Figure 3: Average Performance of Supervised and Few-shot Learning on Groups $\mathbb{G}_1$ and $\mathbb{G}_2$.

training. For example, in the $k$-shot experiment with the PROTEIN dataset from $\mathbb{G}_1$, we trained the model on the other four datasets in this group and then tuned it using $k$ samples from each class in PROTEIN. We repeated this process 10 times and reported the results for $\mathbb{G}_1$ in Tables 10 and 9, and for $\mathbb{G}_2$ in Tables 11 and 12. The average results across all datasets are presented in Figure 3 (b) and (c). Both GraphProp and OFA perform well on $\mathbb{G}_1$, but GraphProp significantly outperforms OFA on $\mathbb{G}_2$. This highlights the key contribution of our paper: **GraphProp achieves both node feature and structural cross-domain generalization**, while previous in-context learning methods primarily focus on node feature generalization and may struggle with datasets lacking node features.

### 5.4 ADDITIONAL EXPERIMENTS IN THE APPENDIX

Additional experiment results are provided in the Appendix. Appendix C.2 compares the structural GFM $f$ with other unsupervised methods like InfoGraph Sun et al. (2019). Appendix C.3 explores cross-domain and random data augmentation to improve GFM $f$ on unseen datasets. Finally, Appendix C.4 presents an ablation study analyzing the contributions of each part of GraphProp.

## 6 CONCLUSION

This paper introduces a new method called GraphProp for training GFMs by predicting graph properties. The idea comes from the observation that graph structures share common properties across different domains, offering more consistent cross-domain information than domain-specific node features. By training a structural GFM, we can improve the structural generalization of other GFMs. Finally, we combine the structural GFM with widely used in-context GFMs to achieve better generalization in both graph structure and node features.

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

# A  APPENDIX

You may include other additional sections here.

# B  RELATED WORKS

This section outlines related works relevant to our study.

## B.1  INTRODUCE TO GRAPH PROPERTIES

We discuss the range of graph properties utilized in our GraphProp method. These properties are detailed in several tables:

- Polynomial computation complexity properties in Table 4.
- NP-hard computation complexity properties in Table 5 and Table 6.
- Node-level properties in Table 7.
- Node pairwise properties in Table 8.

These properties encompass a broad spectrum of graph theory areas, including mathematics, combinatorics, topology, molecular geometry, and computational biochemistry. Our GraphProp method integrates these properties into GFM training, bridging the gap between GFMs and graph theory. Beyond predicting graph properties as discussed in the main paper, we extend GraphProp to train GFMs using both node-level and node pairwise properties. Node-level properties, such as degree or centrality, are represented as vectors $\mathbf{q} = [q_1, ..., q_n] \in \mathbb{R}^n$, where $q_i$ denotes the property of node $i$. These properties are predicted using a node-level regressor:

$$\hat{\mathbf{q}}_{\Theta, \Psi} = \varphi_{\text{node}}(\mathbf{Z}_\Theta; \Psi), \quad \text{where} \quad \mathbf{Z}_\Theta = f(\mathbf{B}; \Theta). \tag{14}$$

Similarly, node pairwise properties such as connectivity or shortest path between nodes $i$ and $j$ are represented by $\mathbf{Q}$, and are predicted using:

$$\hat{\mathbf{Q}}_{\Theta, \Psi} = \varphi_{\text{node-pair}}(\mathbf{Z}_\Theta; \Psi), \quad \text{where} \quad \mathbf{Z}_\Theta = f(\mathbf{B}; \Theta). \tag{15}$$

These diverse and extensive graph theory properties allow GraphProp to learn comprehensive unified graph representations, denoted as $\mathbf{Z}$. Since most graph properties can be incorporated into GraphProp, our approach is fundamental and versatile. In experiments, to ensure manageable computation times, we primarily focus on the fifteen polynomial-time computable graph properties listed in Table 4.

## B.2  GRAPH FOUNDATION MODELS (GFMs)

In the introduction, we categorized GFMs into three types: domain-specific, task-specific, and primitive, providing examples of each (Mao et al.). This section takes a different angle by focusing on GFMs based on LLMs Li et al. (2023b). These can be further classified into two categories depending on their interaction with LLMs: using LLMs as encoders or predictors. Finally, we touch upon in-context learning, which intersects with the LLM-based GFMs discussion.

**LLMs as Encoders** In Text-Attributed Graphs (TAG) of $G^{(m)}$, let $\mathbf{E}^{(m)} \in \mathbb{R}^{n \times d}$ represent the unified node feature matrix and $\mathbf{H} \in \mathbb{R}^{n \times d}$ the hidden graph representation for downstream tasks. The role of LLMs as encoders in GFMs can be summarized with:

$$\mathbf{H} = \text{GNN}(\mathbf{A}^{(m)}, \mathbf{E}^{(m)}) \quad \text{and} \quad \mathbf{E}^{(m)} = \text{LLM}(\text{TAG of } G^{(m)}). \tag{16}$$

For instance, GIANT Chien et al. (2021) enhances a language model with XR-Transformers for multi-label classification and link prediction. SimTeG Duan et al. (2023) and TouchUp-G Zhu et al. (2024) use link prediction methods to refine language models for better structural recognition, with TouchUp-G employing negative sampling and SimTeG using efficient tuning. G-Prompt Huang et al. (2023) adds a graph adapter to language models for node-specific feature extraction, using task-specific prompts for different applications. WalkLM Tan et al. (2024) creates textual sequences from

Table 4: Graph Properties with Polynomial Complexity Computation. Note that it is possible to reduces some of these complexities using techniques such as truncated SVD.

| Name | Notation | Reference | Description | Complexity |
|---|---|---|---|---|
| Size | m | - | Number of edges in graph G | $\mathcal{O}(1)$ |
| Order | n | - | Number of vertices in graph G | $\mathcal{O}(1)$ |
| Fiedler value | - | Fiedler (1973) | Second-smallest eigenvalue of Laplacian matrix | $\mathcal{O}(n^3)$ |
| Diameter | d | Bouttier et al. (2003) | Max eccentricity of any vertex | $\mathcal{O}(nm)$ |
| Estrada index | $EE(G)$ | Estrada (2000) | Measure of protein folding | $\mathcal{O}(n^3)$ |
| Fractional chromatic number | $\chi_f(G)$ | Scheinerman & Ullman (2013) | Smallest k with distribution over independent sets | depends on $G$ |
| Hyper-Wiener index | $WW(G)$ | Randić (1993) | Topological index based on distances | $\mathcal{O}(n^3)$ |
| Lovász number | $\vartheta(G)$ | Lovász (1979) | Upper bound on Shannon capacity of graph | $\mathcal{O}(n^3)$ |
| Parry–Sullivan invariant | $PS(G)$ | Parry & Sullivan (1975) | Quantity of incidence matrices, $PS(G) = \det(\mathbf{I} - \mathbf{A})$ | $\mathcal{O}(n^3)$ |
| Radius | r | Bouttier et al. (2003) | Min eccentricity of any vertex | $\mathcal{O}(nm)$ |
| Randić index | - | Randic (1975) | Sum of $1/(d_i d_j)^{\frac{1}{2}}$ for vertices $i$ and $j$ | $\mathcal{O}(m)$ |
| Rank | - | - | Rank of adjacency matrix | $\mathcal{O}(n^3)$ |
| Splittance | $\sigma(G)$ | Hammer & Simeone (1981) | Measure of distance from a split graph | depends on $G$ |
| Strength | - | Cunningham (1985) | Min ratio of edges removed to components created | depends on $G$ |
| Wiener index | - | Rouvray (2002) | Sum of shortest paths between all vertex pairs | $\mathcal{O}(n^3)$ |

random graph walks and refines a language model to extract useful data representations. METERN Jin et al. (2023) incorporates special tokens to highlight relationship-specific features using a single encoder for shared traits across relationships. LEADING Xue et al. (2023) optimizes the refinement of language models to transfer risk knowledge to graph neural networks with lower computational demands and memory use.

Moreover, some studies explain the node features $\mathbf{E}^{(m)}$ generated by LLMs. TAPE He et al. (2023), for example, prompts LLMs to provide explanations and pseudo labels, enriching the textual data for subsequent fine-tuning of smaller language models into initial node embeddings. Chen et al. (2024) utilize LLMs in graph learning by generating knowledge entities and textual descriptions, which are then processed by PLMs and sentence embedding models. LLM4Mol Qian et al. (2023) employs LLMs for molecular property predictions, generating comprehensive molecular descriptions for further refinement. LLMRec Wei et al. (2024) leverages LLMs to address data scarcity and quality issues in recommendation systems, enhancing interactions and generating additional information for users and items.

**LLMs as Predictors**  LLMs can serve as predictors by translating graph structures into text sequences for direct processing. The flatten-based prediction process first transforms the graph into a sequence of nodes or tokens, $\mathbf{h}$, using the function $\mathrm{Flat}(\cdot)$. Then, it extracts the predicted label through a parsing function, $\mathrm{Parse}(\cdot)$. Thus the flatten-based prediction process is as follows

$$\hat{y} = \mathrm{Parse}(\mathrm{LLM}(\mathbf{h}, t)), \quad \mathbf{h} = \mathrm{Flat}(\mathrm{GNN}(\mathbf{A}, \mathbf{X})) \tag{17}$$

Here, $t$ specifies the prompt for the graph task. For example, GraphText Zhao et al. (2023b) uses graph-syntax trees to transform graph structures into node sequences for training-free reasoning with LLMs. ReLM Shi et al. (2023) employs SMILES strings to linearize molecular structures. GraphTMI Das et al. (2023) integrates graph data with LLMs using motifs and images. GPT4Graph Guo et al. (2023) mimics GNN aggregation to enhance structural input. GIMLET Zhao et al. (2023a)

uses distance-based embeddings to improve LLMs' graph perception, while InstructGLM Ye et al. (2023) uses scalable prompts to optimize the understanding of graph connectivity.

For GNN-based prediction, GNNs analyze graph structures by recursively exchanging and aggregating node information, and integrating these features with LLMs to enhance structural awareness:

$$\hat{y} = \text{Parse}(\text{LLM}(\mathbf{H}, t)), \quad \mathbf{H} = \text{GNN}(\mathbf{A}, \mathbf{X}) \tag{18}$$

where $\mathbf{X}$ is the node embedding matrix, $\mathbf{A}$ the adjacency matrix, and $\mathbf{H}$ the structure-aware embeddings. This approach aligns GNN structural patterns with LLM contextual information, requiring specific tuning to standardize LLM outputs during training. To integrate GNNs' structural patterns with LLMs' contextual abilities, several methods have been developed. GIT-Mol Liu et al. (2024) and MolCA Liu et al. (2023c) use BLIP-2's QFormer Li et al. (2023a) as a cross-modal projector linking graph encoder outputs to LLM inputs. GraphLLM Chai et al. (2023) applies linear projection in prefix tuning to optimize graph prefixes for better integration with graph transformers and LLMs. Similarly, GraphGPT Tang et al. (2024a) and InstructMol Cao et al. (2023) use a simple linear layer for aligning graph data with LLM text processing. DGTL Qin et al. (2023) incorporates disentangled graph embeddings directly into the LLM, enhancing the perception of graph topology and semantics.

**In-context learning** In-context learning is an effective approach for generating unified graph representations across different domains. This method involves describing node features with text, including domain-specific details, and utilizing a large language model (LLM) to create these unified representations. It is particularly relevant to Text-Attributed Graphs (TAG), which use descriptive text prompts to represent the node feature matrix $\mathbf{X}$ of a graph $G$. For a graph $G^{(m)}$ from domain $m$, in-context learning helps create a unified node feature representation $\mathbf{E}^{(m)} \in \mathbb{R}^{n \times d}$, where each row corresponds to the features of a node. These features, along with the adjacency matrix $\mathbf{A}^{(m)}$, are then used by a GNN for downstream tasks such as classification:

$$\hat{y}^{(m)} = \text{GNN}(\mathbf{A}^{(m)}, \mathbf{E}^{(m)}) \quad and \quad \mathbf{E}^{(m)} = \text{LLM}(\text{TAG of } G^{(m)}). \tag{19}$$

Examples include OFA (Liu et al., 2023a), which offers a general solution for building and training foundational GNN models with in-context learning capabilities across various domains. Another example, PRODIGY (Huang et al., 2024), is a pretraining framework designed to facilitate in-context learning on graphs by tailoring both model architecture and pretraining objectives for prompt-based graph tasks. This enables the model to handle a broad spectrum of tasks and graphs straight out-of-the-box. Comprehensive surveys on in-context and prompt learning in graph contexts are available in (Sun et al., 2023; Liu et al., 2023b).

### B.3 GRAPH REASONING METHODS

The graph reasoning methods focus only on reasoning about the graph's structure without involving node features, similar to our approach with graph properties prediction. For example, GraphQA (Fatemi et al., 2023) describes graph connectivity in text and then poses reasoning questions to LLMs, while GraphToken Perozzi et al. (2024) trains LLMs to reason about graph structure with questions like "Is there a cycle in the graph?". CLRS Ibarz et al. (2022) trains LLMs to reason about graph algorithms such as search and greedy algorithms. These methods mainly train LLMs by asking questions and focus on reasoning skills. In contrast, our method, GraphProp, focuses on learning a comprehensive structural representation. The goals are different. However, some graph reasoning tasks, like finding the shortest path, can be incorporated into GraphProp, as it involves predicting node pairwise properties, as shown in Table 8.

### B.4 GRAPH DATA AUGMENTATION

Graph data augmentation (Ding et al., 2022) enhances model performance and generalization by adding new training data. We introduce a graph mixup augmentation technique based on graph matching. Given two graphs $G_1$ and $G_2$ with adjacency matrices $\mathbf{A}_1$ and $\mathbf{A}_2$, and labels $y_1$ and $y_2$, the optimal matching permutation matrix $\mathbf{P} \in \{0, 1\}^{n \times n}$ is obtained by solving

$$\mathbf{P}^* = \underset{\mathbf{P}}{\arg\min} \|\mathbf{A}_1 - \mathbf{P}\mathbf{A}_2\mathbf{P}^\top\|_F^2 \tag{20}$$

The mixup function $\text{mixup}(\cdot, \cdot)$ generates an augmented graph $\hat{G}$ with adjacency matrix $\hat{\mathbf{A}}$ as

$$\hat{\mathbf{A}} = \text{mixup}(\mathbf{A}_1, \mathbf{A}_2) = \sigma\left(\lambda\mathbf{A}_1 + (1-\lambda)\mathbf{P}^*\mathbf{A}_2\mathbf{P}^{*\top} + \epsilon\right), \qquad (21)$$

where $\sigma$ is an activation function mapping to $\{0,1\}^{n \times n}$, $\lambda \in [0,1]$ is a mixup coefficient, and $\epsilon \sim \mathcal{N}(0,1)$ is Gaussian noise. In the G-Mixup method (Han et al., 2022), if $G_1$ and $G_2$ are from the same domain, a soft label $\hat{y}$ is created for downstream tasks as follows

$$\hat{y} = \lambda y_1 + (1-\lambda)y_2 \qquad (22)$$

However, this approach only works when the graphs are from the same domain, as combining labels from different domains is meaningless.

### B.5 GRAPH TRANSFORMER

In addition to the absolute positional encoding graph transformer discussed in the main paper, there is a variant known as the Relative Positional Encoding (RPE) graph transformer, outlined below:

**Definition B.1** (RPE-Graph Transformer). The RPE-Graph Transformer assigns an encoding vector to each pair of nodes in graph $G$ and then reflects this encoding to a value, like the shortest path between nodes Ying et al. (2021). We define a mapping function $\phi' : \mathbb{R}^{n \times n} \to \mathbb{R}^{n \times n}$, which computes the RPE matrix $\mathbf{B}' = \phi'(\mathbf{A})$ with $\mathbf{B}' \in \mathbb{R}^{n \times n}$. The self-attention in this transformer is modified as follows:

$$\begin{aligned} \text{Adding:} \quad & \text{attn}(\mathbf{H}) = \text{softmax}\left(\frac{\mathbf{Q}\mathbf{K}^\top}{\sqrt{d_K}} + \mathbf{B}'\right)\mathbf{V}, \\ \text{or Hadamard Product:} \quad & \text{attn}(\mathbf{H}) = \text{softmax}\left(\frac{\mathbf{Q}\mathbf{K}^\top}{\sqrt{d_K}} \odot \mathbf{B}'\right)\mathbf{V}. \end{aligned} \qquad (23)$$

Several examples of graph transformers illustrate the diversity in their applications and methodologies. GraphBert Zhang et al. (2020) encodes nodes using the graph structure without changing the fundamental attention mechanism. Gophormer Zhao et al. (2021) introduces scalability through sampling techniques, while NAGphormer Chen et al. (2022) and Nodeformer Wu et al. (2022) are node-level transformers utilizing kernelized attention mechanisms. Difformer Wu et al. (2023) operates on a continuous-time, diffusion-based model. GraphGPS Rampášek et al. (2022) combines message-passing networks with attention mechanisms, allowing for a variety of embeddings. Graphormer Ying et al. (2021) integrates dense attention with structural features like centrality and spatial encodings, while GraphiT Mialon et al. (2021) incorporates relative positional encodings based on diffusion kernels. Finally, BigBird Zaheer et al. (2020) introduces sparse transformer models for better performance and scalability.

### B.6 OTHER GRAPH THEORY BENCHMARKS

To learn comprehensive unified representations across domains, our GraphProp incorporates a broad set of graph properties that can be computed in polynomial time, including some properties like the fractional chromatic number and graph strength, which are introduced to graph learning for the first time. Existing benchmarks for graph properties are not well-suited for training GFMs. For example, the GNN benchmarking dataset Dwivedi et al. (2023) includes only three properties—connectivity, diameter, and spectral radius—limiting its scope for comprehensive graph representation learning. Similarly, the Circular Skip Link (CSL) dataset Murphy et al. (2019) is too small, with only 150 graphs. Other benchmarks, such as GraphQA Fatemi et al. (2023), CLRS Veličković et al. (2022), and NLGraph Wang et al. (2024), focus on reasoning tasks like shortest paths and connectivity, often converting graphs into text for LLMs. These benchmarks are designed to train reasoning abilities rather than to provide comprehensive graph property learning.

## C ADDITIONAL EXPERIMENT DETAILS

In this section, we provide additional experimental details and some numerical results.

## C.1 Numerical Results of Few-Shot Learning

Here, we present the numerical results of few-shot learning, as shown in Table 10, Table 9, Table 11, and Table 12.

## C.2 Unsupervised Graph Representation Learning

The structural GFM $F$ can be used as an unsupervised graph representation learning model, and we compare it with other unsupervised methods like InfoGraph Sun et al. (2019), GCL You et al. (2020), and GraphACL (Luo et al., 2023). We evaluate the models based on clustering performance, using clustering accuracy (ACC) and Normalized Mutual Information (NMI) as metrics. The experiments are conducted on group $\mathbb{G}_2$, focusing on graphs without node features. The results are reported in Table 13.

## C.3 Data Augmentation

For group $\mathbb{G}_2$, we added either 500 cross-domain augmented data or 500 randomly generated data to train the structural GFM $f$. The classification results, shown in Table 14, demonstrate that data augmentation improves performance.

## C.4 Ablation Study

In this section, we analyse GraphProp by removing each part of it.

**Removing In-context Learning From GraphProp:** When in-context learning is removed from the GraphProp framework, it becomes a graph transformer with $\mathbf{Z}$ as its positional encoding. We compare this version with other graph transformers in graph classification tasks, and the results are shown in Table 15.

**Removing Structural GFM From GraphProp** When the structural GFM is removed from the GraphProp framework, the remaining part is identical to the in-context learning OFA Liu et al. (2023a). We compare this version with OFA in graph classification tasks, and the results are shown in Table 16.

**Removing Some Graph Properties From GraphProp** In the main paper, we used fifteen graph properties listed in Table 4. Now, we randomly remove some of these properties and repeat the supervised learning experiments. For each number of removed properties, we repeat the process ten times and report the average performance. The results are shown in Table 17.

Table 5: Graph Properties with NP-hard Complexity Computation (Part I)

| Name | Notation | Reference | Description |
|---|---|---|---|
| Arboricity | - | Edmonds (1965) | Min. number of forests into which edges can be partitioned |
| Biclique Cover Number | $d(G)$ | Amilhastre et al. (1997) | Min. number of bicliques of $G$ |
| Boxicity | $box(G)$ | Chandran et al. (2010) | Min. dimension for $G$ as an intersection graph of axis-parallel boxes |
| Carving Width | - | Seymour & Thomas (1994) | Edges separating clusters in a hierarchical clustering of vertices |
| Cheeger Constant | $h(G)$ | Mohar (1989) | Numerical measure of a graph's "bottleneck" |
| Chromatic Number | $\chi(G)$ | Jensen & Toft (2011) | Smallest number of colors needed to color $G$ |
| Chromatic Index | $\chi'(G)$ | Akiyama et al. (1980) | Smallest number of colors needed in a proper edge coloring of $G$ |
| Clique Number | $\omega(G)$ | Alba (1973) | Number of vertices in a maximum clique in $G$ |
| Colin de Verdière's Invariant | $\mu(G)$ | de Verdiere (1990) | Max. multiplicity of the 2nd eigenvalue of certain Schrödinger operators |
| Conductance | $\varphi(G)$ | Jerrum & Sinclair (1988) | Parameter tied to mixing time of a Markov chain, analyzing random walk convergence |
| Cop Number | - | Bonato (2011) | Min. number of cops to ensure a win in a pursuit–evasion game on the graph |
| Crossing Number | $cr(G)$ | Purchase et al. (1996) | Lowest number of edge crossings in a plane drawing of $G$ |
| Dimension | - | Erdös et al. (1965) | Least integer $n$ for classical representation of $G$ in Euclidean space with edges of unit length |
| Dissociation Number | $diss(G)$ | Yannakakis (1981) | Number of vertices in a max. cardinality dissociation set in $G$ |
| Distinguishing Number | - | Albertson & Collins (1996) | Min. number of colors in a distinguishing coloring |
| Domatic Number | - | Cockayne & Hedetniemi (1975) | Max. size of a domatic partition |
| Domination Number | $\gamma(G)$ | Alber et al. (2004) | Number of vertices in a smallest dominating set for $G$ |
| Edge Covering Number | $\rho(G)$ | Lewis (1983) | Size of a minimum edge covering |
| Entanglement | - | Berwanger & Grädel (2005) | Measure of how strongly cycles of $G$ are intertwined |
| Friendly Index | $FI(G)$ | Kwong et al. (2008) | Absolute value of the difference between the number of edges labeled 0 and 1 |
| Girth | - | Diestel (2024) | Length of the shortest cycle in the graph |
| Graph Bandwidth | - | Chinn et al. (1982) | Minimal bandwidth of a symmetric matrix which is an adjacency matrix of $G$ |
| Pebbling Number | $\pi(G)$ | Chung (1989) | Lowest natural number satisfying pebbling game conditions |
| Toughness | - | Bauer et al. (2006) | Max. t for which $G$ is t-tough |
| Grundy Number | - | Erdös et al. (2003) | Max. number of colors in a greedy coloring strategy |
| Hadwiger Number | - | Bollobás et al. (1980) | Size of the largest complete graph obtained by contracting edges of $G$ |
| Hosoya Index | - | Hosoya (1971) | Total number of matchings in $G$ |
| Independence Number | $\alpha(G)$ | Godsil & Royle (2001) | Size of maximum independent set of $G$ |
| Intersection Number | - | Gross et al. (2018) | Smallest number of elements in a representation of $G$ as an intersection graph |
| Linear Arboricity | - | Akiyama et al. (1981) | Smallest number of linear forests its edges can be partitioned into |
| Matching Number | $\nu(G)$ | Gibbons (1985) | Size of a maximum matching |
| Matching Preclusion Number | $mp(G)$ | Brigham et al. (2005) | Min. number of edges whose deletion eliminates all perfect matchings |

Table 6: Graph Properties with NP-hard Complexity Computation (Part II)

| Name | Notation | Reference | Description |
|---|---|---|---|
| Meshedness Coefficient | - | Buhl et al. (2004) | Invariant of planar graphs measuring the number of bounded faces |
| Metric Dimension | - | Feng et al. (2013) | Min. cardinality of a vertex subset such that all other vertices are uniquely determined by their distances to this subset |
| Minimum Rank | $mr(G)$ | Fallat & Hogben (2007) | Smallest rank of any generalized adjacency matrix of $G$ |
| Padmakar–Ivan Index | $PI(G)$ | Khadikar et al. (2001) | Sum over all edges $uv$ of $G$ of the number of edges not equidistant from $u$ and $v$ |
| Pathwidth | - | Diestel & Kühn (2005) | Measure of how much the path was thickened to form $G$ |
| Perron Number | - | Borwein (2002) | Algebraic integer greater than 1 with all conjugate elements smaller in absolute value |
| Queue Number | $qm(G)$ | Heath & Rosenberg (1992) | Min. number of queues in a queue layout |
| Shannon Capacity | - | Lovász (1979) | Number of independent sets of strong graph products |
| Slope Number | - | Pach & Pálvölgyi (2006) | Min. number of distinct slopes of edges in a drawing of $G$ |
| Szeged Index | $Sz(G)$ | Gutman (1994) | Topological index of a molecule, generalizes the Wiener index |
| Thickness | - | Beineke & Harary (1965) | Min. number of planar graphs into which the edges of $G$ can be partitioned |
| Thue Number | - | Alon et al. (2002) | Variation of chromatic index used to study square-free words |
| Treewidth | - | Diestel (2024) | Integer specifying how far $G$ is from being a tree |
| Twin Width | - | Bonnet et al. (2021) | Number associated with $G$, used to study parameterized complexity of algorithms |
| Vertex Connectivity | - | Schrijver et al. (2003) | Largest $k$ for which the graph is $k$-vertex-connected |
| Vertex Cover Number | $\tau$ | Chen et al. (2006) | Size of a minimum vertex cover |

Table 7: Node-level Properties

| Name | Notation | Reference | Description | Complexity |
|---|---|---|---|---|
| Betweenness Centrality | - | Freeman (1977) | Centrality based on shortest paths between nodes | $\mathcal{O}(n^3)$ |
| Closeness Centrality | $C_B(x)$ | Sabidussi (1966) | Centrality based on inverse of the total distance to all other nodes | $\mathcal{O}((n+m)n)$ |
| Degree | - | - | Number of connections a node has | $\mathcal{O}(1)$ |
| Degree Distribution | - | - | Distribution of node degrees in the network | $\mathcal{O}(n)$ |
| Katz Centrality | - | Katz (1953) | Centrality measuring a node's influence through connections | $\mathcal{O}(n^3)$ |

Table 8: Node Pairwise Properties

| Name | Notation | Reference | Description | Complexity |
|---|---|---|---|---|
| Connectivity | - | - | Predict if two nodes are connected by an edge | $\mathcal{O}(1)$ |
| Shortest Path | - | Yu & Yang (1998) | Predict the shortest path between two nodes | $\mathcal{O}(n^3)$ |
| Maximum Flow | - | Schrijver (2002) | Predict the maximum flow between two nodes in a weighted graph | $\mathcal{O}(n^2m)$ |

Table 9: Results of few-shot learning on PROTEIN and NCI1 data in group $\mathbb{G}_1$.

| Data | PROTEINS | | | | NCI1 | | | |
|---|---|---|---|---|---|---|---|---|
| task | 10-shot | 5-shot | 1-shot | 0-shot | 10-shot | 5-shot | 1-shot | 0-shot |
| OFA-st | 70.04 ± 4.41 | 63.45 ± 7.01 | 57.11 ± 5.10 | 48.60 ± 4.40 | 61.52 ± 6.19 | 56.33 ± 5.63 | 51.83 ± 7.82 | 50.48 ± 6.05 |
| OFA-e5 | 68.27 ± 5.88 | 60.50 ± 6.98 | 55.22 ± 4.25 | 51.41 ± 5.59 | 64.83 ± 4.20 | 55.64 ± 9.72 | 48.91 ± 3.74 | 54.82 ± 7.92 |
| OFA-llama2 | 65.32 ± 4.92 | 62.73 ± 2.29 | 56.37 ± 3.52 | 55.92 ± 7.92 | 62.70 ± 7.35 | 54.78 ± 3.90 | 50.18 ± 2.45 | 49.50 ± 2.37 |
| GraphProp-st | 67.64 ± 6.05 | 61.34 ± 4.74 | 52.83 ± 6.67 | 54.34 ± 6.28 | 60.41 ± 3.44 | 57.19 ± 2.26 | 53.35 ± 6.36 | 43.79 ± 4.72 |
| GraphProp-e5 | 75.88 ± 3.93 | 66.60 ± 8.63 | 54.66 ± 7.44 | 51.65 ± 8.74 | 63.97 ± 5.73 | 60.02 ± 8.37 | 55.72 ± 5.40 | 45.45 ± 8.58 |
| GraphProp-llama2 | 71.24 ± 5.10 | 64.91 ± 1.37 | 59.24 ± 4.39 | 53.28 ± 4.37 | 67.25 ± 4.87 | 59.91 ± 7.63 | 54.48 ± 6.23 | 51.13 ± 4.61 |

Table 10: Results of few-shot learning on AIDS and HIV data in group $\mathbb{G}_1$.

| Data | AIDS | | | | HIV | | | |
|---|---|---|---|---|---|---|---|---|
| task | 10-shot | 5-shot | 1-shot | 0-shot | 10-shot | 5-shot | 1-shot | 0-shot |
| OFA-st | 53.71±7.32 | 46.74±7.43 | 47.50±6.77 | 41.81±8.27 | 68.16±6.31 | 60.53±5.61 | 50.08±7.33 | 47.76±6.37 |
| OFA-e5 | 56.95±2.76 | 54.96±4.82 | 49.68±2.03 | 40.62±7.94 | 62.37±3.66 | 57.96±7.74 | 52.34±5.19 | 53.36±2.60 |
| OFA-llama2 | 58.42±4.18 | 49.42±2.28 | 46.42±7.94 | 35.46±3.67 | 65.78±8.32 | 62.25±6.88 | 54.23±3.61 | 43.45±5.41 |
| GraphProp-st | 59.63±7.45 | 47.23±6.97 | 42.27±8.26 | 42.37±6.79 | 67.34±5.23 | 55.43±3.97 | 51.91±4.48 | 46.38±3.32 |
| GraphProp-e5 | 56.28±3.24 | 55.60±5.21 | 48.45±9.37 | 37.93±3.80 | 61.49±4.75 | 57.51±4.05 | 56.70±6.34 | 47.82±4.14 |
| GraphProp-llama2 | 52.07±6.98 | 50.81±4.76 | 42.38±6.40 | 44.04±9.03 | 68.60±8.17 | 54.60±9.26 | 53.66±9.85 | 53.71±9.85 |

Table 11: Results of few-shot learning on COLLAB and REDDIT-B data in group $\mathbb{G}_2$.

| Data | COLLAB | | | | REDDIT-B | | | |
|---|---|---|---|---|---|---|---|---|
| task | 10-shot | 5-shot | 1-shot | 0-shot | 10-shot | 5-shot | 1-shot | 0-shot |
| OFA-st | 54.16 ± 7.46 | 48.51 ± 8.51 | 45.80 ± 7.36 | 37.34 ± 4.06 | 58.14 ± 5.12 | 53.15 ± 2.29 | 51.32 ± 6.54 | 45.61 ± 4.72 |
| OFA-e5 | 52.22 ± 9.51 | 41.69 ± 5.45 | 37.62 ± 5.22 | 32.62 ± 7.49 | 43.95 ± 4.34 | 51.90 ± 6.30 | 56.91 ± 3.13 | 52.98 ± 7.85 |
| OFA-llama2 | 51.04 ± 2.87 | 46.47 ± 6.89 | 40.27 ± 6.87 | 45.53 ± 5.30 | 55.63 ± 8.68 | 48.64 ± 5.75 | 52.16 ± 6.27 | 54.27 ± 2.62 |
| GraphProp-st | 65.19 ± 4.24 | 63.92 ± 7.12 | 48.45 ± 4.79 | 39.87 ± 4.84 | 59.19 ± 5.41 | 57.11 ± 7.17 | 45.62 ± 4.38 | 49.44 ± 6.97 |
| GraphProp-e5 | 57.35 ± 8.90 | 58.03 ± 5.27 | 50.18 ± 8.51 | 46.34 ± 8.73 | 61.08 ± 3.19 | 55.47 ± 4.98 | 54.24 ± 8.76 | 52.32 ± 7.22 |
| GraphProp-llama2 | 60.81 ± 5.13 | 59.25 ± 4.76 | 47.90 ± 5.14 | 42.91 ± 6.15 | 64.63 ± 7.63 | 54.26 ± 6.33 | 51.77 ± 7.21 | 47.75 ± 3.60 |

Table 12: Results of few-shot learning on IMDB-B and DD data in group $\mathbb{G}_1$.

| Data | IMDB-B | | | | DD | | | |
|---|---|---|---|---|---|---|---|---|
| task | 10-shot | 5-shot | 1-shot | 0-shot | 10-shot | 5-shot | 1-shot | 0-shot |
| OFA-st | 57.21 ± 6.71 | 55.39 ± 4.65 | 54.36 ± 5.84 | 56.21 ± 7.56 | 63.81 ± 7.77 | 59.11 ± 4.24 | 54.29 ± 8.65 | 51.71 ± 4.16 |
| OFA-e5 | 58.49 ± 7.74 | 53.27 ± 3.74 | 51.26 ± 6.35 | 51.38 ± 6.15 | 60.21 ± 4.66 | 54.63 ± 6.75 | 53.12 ± 4.31 | 54.66 ± 5.77 |
| OFA-llama2 | 55.57 ± 2.75 | 51.84 ± 2.47 | 48.71 ± 3.49 | 50.73 ± 4.24 | 64.85 ± 2.21 | 57.39 ± 8.67 | 51.88 ± 2.38 | 55.93 ± 8.04 |
| GraphProp-st | 65.39 ± 3.28 | 61.49 ± 7.38 | 49.36 ± 4.82 | 53.88 ± 5.73 | 73.22 ± 6.49 | 61.48 ± 5.18 | 52.47 ± 6.25 | 52.74 ± 5.65 |
| GraphProp-e5 | 72.37 ± 5.27 | 65.81 ± 5.42 | 57.57 ± 7.50 | 53.45 ± 3.48 | 76.95 ± 7.23 | 63.27 ± 4.39 | 58.75 ± 1.76 | 51.15 ± 7.53 |
| GraphProp-llama2 | 69.36 ± 4.36 | 67.52 ± 8.59 | 62.57 ± 2.37 | 51.26 ± 8.20 | 75.16 ± 5.81 | 64.71 ± 8.24 | 57.26 ± 7.31 | 55.78 ± 8.43 |

Table 13: ACC and NMI of Graph Clustering on datasets in group $\mathbb{G}_2$.

| Method | Metric | DD | COLLAB | IMDB-B | REDDIT-B | REDDIT-M5K |
|---|---|---|---|---|---|---|
| InfoGraph | ACC | 0.57 ± 0.06 | 0.58 ±0.28 | 0.67 ±0.07 | 0.57 ±0.07 | 0.58 ±0.09 |
| | NMI | 0.24 ± 0.04 | 0.37 ±0.01 | 0.18 ±0.05 | 0.22 ±0.04 | 0.25 ±0.03 |
| GCL | ACC | 0.59 ± 0.01 | 0.53 ±0.12 | 0.61 ±0.04 | 0.56 ±0.01 | 0.50 ±0.18 |
| | NMI | 0.23 ± 0.02 | 0.27 ± 0.03 | 0.21 ±0.05 | 0.14 ±0.03 | 0.27 ±0.05 |
| GraphACL | ACC | 0.59 ± 0.03 | 0.56 ±0.05 | 0.60 ±0.03 | 0.57 ±0.07 | 0.56 ±0.04 |
| | NMI | 0.32 ± 0.03 | 0.29 ±0.06 | 0.33 ±0.30 | 0.23 ± 0.03 | 0.22 ± 0.09 |
| GraphProp | ACC | 0.58 ± 0.05 | 0.61 ±0.18 | 0.62 ±0.02 | 0.64 ±0.04 | 0.61 ±0.03 |
| | NMI | 0.36 ± 0.02 | 0.33 ±0.09 | 0.34 ±0.01 | 0.28 ± 0.02 | 0.27 ±0.01 |

Table 14: Data Augmentation (Adding 500 new graphs). Results of supervised learning.

| Data | Augmentation | COLLAB | IMDB-B | DD | REDDIT-B | REDDIT-M5K |
|---|---|---|---|---|---|---|
| Metric | | ACC ↑ | ACC↑ | ACC↑ | ACC↑ | ACC↑ |
| GraphProp-e5 | no | 81.35 ± 1.32 | 82.78 ± 1.85 | 82.31 ± 1.41 | 85.32 ± 1.17 | 59.36 ± 1.27 |
| GraphProp-e5 | cross-domain | 86.24 ± 1.29 | 86.15 ± 1.24 | 86.80 ± 1.30 | 87.13 ± 1.26 | 62.37 ± 1.18 |
| GraphProp-e5 | random | 87.16 ± 1.15 | 85.26 ± 1.13 | 84.14 ± 1.18 | 86.11 ± 1.18 | 60.87 ± 1.31 |
| GraphProp-llama2 | no | 82.64 ± 1.58 | 83.42 ± 1.70 | 80.25 ± 1.38 | 84.38 ± 1.26 | 60.93 ± 1.45 |
| GraphProp-llama2 | cross-domain | 85.17 ± 1.27 | 85.11 ± 1.12 | 84.47 ± 1.56 | 88.49 ± 1.29 | 62.31 ± 1.69 |
| GraphProp-llama2 | random | 87.88 ± 1.10 | 86.90 ± 1.78 | 83.51 ± 1.72 | 86.10 ± 1.35 | 61.16 ± 1.57 |

Table 15: Ablation Study: Removing In-context Learning. Supervised learning results.

| Data | COLLAB | IMDB-B | DD | REDDIT-B | REDDIT-M5K |
|---|---|---|---|---|---|
| Metric | ACC ↑ | ACC ↑ | ACC ↑ | ACC ↑ | ACC ↑ |
| GraphGPS | 73.24 ± 1.92 | 75.18 ± 1.54 | 74.12 ± 1.28 | 74.27 ± 1.32 | 58.17 ± 1.12 |
| Graphormer | 71.16 ± 1.27 | 76.27 ± 1.79 | 75.63 ± 1.21 | 75.10 ± 1.49 | 54.66 ± 1.63 |
| GraphProp | 75.22 ± 1.26 | 78.33 ± 1.13 | 77.22 ± 1.49 | 79.61 ± 1.78 | 59.14 ± 1.87 |

Table 16: Ablation Study: Removing Structural GFM. Supervised learning results.

| Data | COLLAB | IMDB-B | DD | REDDIT-B | REDDIT-M5K |
|---|---|---|---|---|---|
| Metric | ACC ↑ | ACC ↑ | ACC ↑ | ACC ↑ | ACC ↑ |
| OFA-st | $74.24 \pm 1.43$ | $75.92 \pm 1.58$ | $77.34 \pm 1.14$ | $80.03 \pm 1.22$ | $53.28 \pm 1.24$ |
| OFA-e5 | $76.25 \pm 1.09$ | $78.19 \pm 1.33$ | $76.65 \pm 1.23$ | $79.62 \pm 1.17$ | $55.17 \pm 1.59$ |
| OFA-llama2 | $75.44 \pm 1.37$ | $77.69 \pm 1.25$ | $75.46 \pm 1.80$ | $78.23 \pm 1.35$ | $54.26 \pm 1.13$ |
| GraphProp-st | $75.15 \pm 1.31$ | $76.71 \pm 1.68$ | $75.61 \pm 1.36$ | $79.66 \pm 1.11$ | $55.90 \pm 1.15$ |
| GraphProp-e5 | $73.66 \pm 1.17$ | $76.21 \pm 1.13$ | $77.28 \pm 1.42$ | $78.13 \pm 1.53$ | $54.23 \pm 1.44$ |
| GraphProp-llama2 | $76.59 \pm 1.65$ | $78.34 \pm 1.17$ | $77.13 \pm 1.01$ | $79.61 \pm 1.54$ | $52.28 \pm 1.78$ |

Table 17: Ablation Study: Removing Graph Properties. Supervised learning results.

| Data | Num. of Properties | COLLAB | IMDB-B | DD | REDDIT-B | REDDIT-M5K |
|---|---|---|---|---|---|---|
| Metric | | ACC ↑ | ACC ↑ | ACC ↑ | ACC ↑ | ACC ↑ |
| GraphProp-llama2 | 15 | $82.64 \pm 1.58$ | $83.42 \pm 1.70$ | $80.25 \pm 1.38$ | $84.38 \pm 1.26$ | $60.93 \pm 1.45$ |
| GraphProp-llama2 | 10 | $74.18 \pm 1.65$ | $78.17 \pm 1.58$ | $74.13 \pm 1.70$ | $78.44 \pm 1.79$ | $55.52 \pm 1.88$ |
| GraphProp-llama2 | 5 | $63.24 \pm 1.17$ | $74.28 \pm 1.32$ | $68.29 \pm 1.15$ | $71.16 \pm 1.14$ | $51.47 \pm 1.91$ |

