# OpenReview forum: "GraphProp: Training the Graph Foundation Models using Graph Properties"
_ICLR.cc/2025/Conference — Submitted to ICLR 2025_

### Official Review · Reviewer_kWMv · 2024-10-24

**Soundness:** 2
**Presentation:** 3
**Contribution:** 2
**Rating:** 5
**Confidence:** 4

**Summary:**

This paper proposes GraphProp for cross-domain graph-level generalization tasks, which emphasizes structural consistent information. GraphProp pre-trains a structure GFM by pre-defined graph properties regression, and generates the structural embedding that is concatenated to node representation for graph classification supervised fine-tuning. The experiments shows its effectiveness compared to conventional in-context learning methods.

**Strengths:**

1. This paper addresses the challenge of cross-domain generalization for graph-level tasks by exploring structural feature consistency across domains, offering a novel perspective for cross-domain generalization learning on graphs.
2. The paper is well-organized, clearly presenting the research motivation and the specific implementation methods.
3. The authors analyze the strengths and weaknesses of the paper, clarifying the applicability and limitations of the proposed method.

**Weaknesses:**

1. The pre-computed graph properties are a set of manually selected, discrete values. The motivation that cross-domain structural features can be extracted through regression of these values requires empirical validation. Additionally, the computational complexity varies significantly with different graph sizes, resulting in limited scalability of the method.
2. According to the formal definition of the method, the authors assume consistency in node counts, input spaces, and output spaces across domains. While using language models to extract node attributes can ensure input space consistency, other assumptions are difficult to satisfy in real-world applications, which limits the method's applicability.
3. The experimental comparisons only include OFA as a baseline method for graph cross-domain generalization. This limited comparison is insufficient to demonstrate the superiority of the proposed method.

**Questions:**

1. Which language model is used in Equation (6), and how are the corresponding structural features and property features extracted?
2. In Figure 1, the authors claim that structural correlations across datasets are stronger than node feature correlations. However, considering C and E in Equation (6), C contains many common connecting words like "connected" and follows a relatively uniform linguistic format. Therefore, features extracted by the LLM would inevitably contain shared semantic features. Meanwhile, in E, due to different node attribute descriptions across datasets, the correlations extracted by the language model are naturally lower. This example does not effectively demonstrate that structural features have better cross-domain correlations.
3. Where is the RAG demonstrated in Figure 2, and what is the source of retrieval?
4. In the Data Augmentation of section 3.2, if two graphs are of different sizes, would this make the data augmentation impossible to perform?

---

### Official Review · Reviewer_yxuV · 2024-10-27

**Soundness:** 2
**Presentation:** 3
**Contribution:** 2
**Rating:** 3
**Confidence:** 4

**Summary:**

This paper propose GraphProp, a method that combines a graph property predictor model with an LLM for graph classification. Specifically, the graph property predictor is a graph transformer pre-trained to predict a series of predefined graph properties. After pre-training, the intermediate node representations of the pre-trained graph transformer are combined with the node representations from an LLM to make the final prediction on a graph. The paper conducted experiments comparing the model with GNN-based model and GNN+LLM approach, and showed better results.

**Strengths:**

The idea of training a universal structural representation is interesting. While most recent approaches try to combine graph learning ability and semantic learning ability, this work shows a promising direction to explore the possibility of training a model that understands graph structure well and injects that information into the semantic model.

Overall, the paper is easy to follow, and it also provides a connection of the work to existing approaches.

**Weaknesses:**

- From the novelty and contribution perspective, while the idea of universal graph representation is interesting and promising, the proposed pipeline to acquire such an ability seems implausible to me. Specifically,
    - Can a graph transformer with the proposed positional encoding predict all proposed graph properties? Note that the graph transformer and the spectral encoding are constrained by 3-wl expressivity, and is their combination (theoretically) capable of predicting all the properties? You should theoretically or empirically justify that the proposed methods can indeed predict the targets.
    - Suppose the model has the expressivity to predict the set of properties, does this learning pipeline suffice to be called a "foundation model?" On one end, you will always need to design new properties to fit the new data. On the other, the current model only tackles graph classification problem, but ideally one would want a foundation model to solve all tasks in a domain.

- From the presentation perspective, the motivation example seems hand-waving to me. The representation for TSG is much more uniform across datasets, and most words are "Node X: Connected to", causing high correlation among datasets. Whereas the representation for TAG can be a lot more diverse as the node description involves atom names, which can differ quite significantly across datasets, leading to lower correlation. However, this difference in semantics does not say much about the transferability of TAG and TSG, it also measures the semantic similarity, which is heavily influenced by how you setup the text description but not by the inherent information. I understand the message you try to convey, yet the example does not really make sense to me. I suggest providing a more rigorous analysis of the transferability not based on the textual description.

- From the experiment perspective, you should consider adding more baselines for a comprehensive comparison, and including several interesting works, such as GIMLET and  LLM4Mol, is still important. It seems like you also only conducted graph-level tasks. Moreover, you should compare your model with a variant where you do not use a pre-trained graph property predictor, but, instead, you can directly concatenate the set of graph properties to the representation you obtained from the LLM. This is particularly important to validate the property prediction pre-training. You should also report how well your property predictor perform, because it also makes sense to use such a predictor when it's doing its job well. Does your method apply to link and node tasks? If so, it would be nice to have those results.

**Questions:**

Please see above.

---

### Official Review · Reviewer_DvaD · 2024-11-03

**Soundness:** 3
**Presentation:** 3
**Contribution:** 3
**Rating:** 6
**Confidence:** 4

**Summary:**

The authors focus on proposing a universal Graph Foundation Model, GraphProp, for graph-level tasks. Specifically, they proposed a structural pre-training strategy that incorporates graph theory to encode common structural knowledge across domains. Additionally, GraphProp leverages large language models to unify the data space of different graph datasets and designs an attention-based encoding strategy for label prediction. Extensive experimental results demonstrate the superiority of the proposed method.

**Strengths:**

S1. The proposed structural pre-training strategy is both intriguing and insightful to me. By pre-training the structural encoder with graph properties that have automatically obtainable labels, the proposed GraphProp enables the model to learn graph universal and hidden patterns.

S2. The paper is well-organized and easy to follow.

S3. The experiments are comprehensive, demonstrating the effectiveness of GraphProp.

**Weaknesses:**

W1. It's not clear how to apply GraphProp to zero-shot scenarios. Although I know that the used baseline (OFA) can be applied to zero-shot, as a reader, I am more curious to see how the proposed model can enhance the model performance in the zero-shot scenario. Especially how to use comprehensive training part in a zero-shot scenario. I suggest the authors provide a detailed explanation for this.

W2. Figure 1 is missing a comparison with noise. From the prompts of TSGs and TAGs, there are more meaningless but similar words (e.g. "connected to" and "and") between TSGs of different graphs. This may be a reason why it seems that Figure 1 (a) has higher similarity than Figure 2 (a). I suggest the authors add meaningless noise prompts for comparison (e.g. TSGs generated by randomly linked graphs) to further prove the point.

W3. There are some writing errors in the paper. For example, Appendix A should be deleted. In addition, although I think the section 2 is well written and makes it easy for the reader to understand the basics, it is too long, and the introduction of the methods and experiments section seems a bit inadequate. Perhaps some of the subsections in section 2 could be combined or bolded instead of being separate subsections, which would make good use of the blank space.

**Questions:**

Please see above.

---

### Official Review · Reviewer_2PTC · 2024-11-04

**Soundness:** 2
**Presentation:** 2
**Contribution:** 2
**Rating:** 3
**Confidence:** 5

**Summary:**

This paper presents _GraphProp_, a proposed GFM by prioritizing structural over node-specific information to improve cross-domain graph-level task performance. _GraphProp_ operates in two stages: first, it trains a structural GFM to predict inherent graph properties. Second, it uses these structural embeddings as positional encodings to train a comprehensive GFM, incorporating domain-specific node features and labels to further generalize across data. Experimental results show _GraphProp_ perform better under specific setting comparing with OFA.

**Strengths:**

1. The paper addresses the GFM problem, which is highly significant across the entire field of graph analysis and presents a considerable challenge. However, the proposed approach is relatively simple given the complexity of the problem.
2. The paper provides a relatively clear related work section in the Appendix, which is helpful for readers outside this niche area to quickly gain foundational understanding.
3. The paper provides a clear explanation and comparison of various graph properties and their computational complexities in the Appendix.

**Weaknesses:**

1. The primary concern with this paper lies in its misalignment between the proposed goal of achieving a GFM and the actual experiments conducted. The current experiments are limited to datasets designed for graph classification, focusing on a single task type, all at the graph level. This approach does not substantiate the broader scope implied by a GFM. We recommend either narrowing the scope explicitly to a GFM designed specifically for graph classification tasks or enhancing the experimental framework by incorporating more diverse graph datasets for pre-training and downstream testing. Additionally, the paper does not provide any explanation as to why this approach could contribute to graph-level tasks. The only related mention occurs at line 362, where it is stated that GraphProp faces difficulties with larger graphs due to its use of computationally complex graph properties. However, this is a matter of methodological design rather than a theoretically justified reason.
2. The observations related to Figure 1 devote substantial space to discussing an intuitively evident point. Specifically, Figure 1 merely illustrates that when only graph structure is present without node information (a), the representations generated by the LLM exhibit low discriminative power across different graph datasets. Conversely, when only node information is present without graph structure (b), the LLM-generated representations display higher discriminative power across these datasets. This is straightforward to understand, as it’s clear from the input text provided to the LLM that the TSG lacks distinctive features, whereas the TAG demonstrates significant discriminative capability.
3. From a methodological perspective, Structural GFM merely trains a model capable of predicting various structural indicators of a graph and applies this model in downstream tasks. However, the critical question arises at best, what the model learns is to predict structural characteristics for any given graph. Why, then, can’t the structural characteristics of downstream graph data be used directly as $Z$, instead of relying on the output of a frozen Structural GFM model, to train the so-called Comprehensive GFM? Theoretically, this approach should perform at least as well as the current method.
4. Modeling graph structure and node features separately and training the structure-related component during pretraining is a common approach, as demonstrated in [1]. More importantly, the paper does not substantiate, either theoretically or empirically, why this particular design of Structural GFM is superior to other self-supervised models that learns the graph structure. Additionally, it remains unclear why the representation $Z$ output by Structural GFM can be directly added to the representation $E$ obtained from the LLM, given that they belong to different representational spaces. If $Z$ is intended to function as a positional encoding (PE), it should be compared with other existing PE methods.
5. As a GFM paper, the experiments are relatively limited, especially in terms of baseline comparisons. There are some papers that could serve as valuable baselines for comparison [2,3,4].
6. In practical application, when node features are available, the paper suggests using TAG. However, even with node features, it is possible to construct TSG as input for the LLM. Why is this option not utilized? Using TSG could potentially provide the LLM with richer information, leading to higher quality embeddings $E$.
7. In the Appendix, none of the results are highlighted to show the best or second-best performance, either through bolding or other visual indicators. This makes it challenging for readers to clearly understand the conclusions conveyed by the other experiments.

[1] [GraphControl: Adding Conditional Control to Universal Graph Pre-trained Models for Graph Domain Transfer Learning](https://arxiv.org/abs/2310.07365)

[2] [THUDM/GraphAlign: GraphAlign: Pretraining One Graph Neural Network on Multiple Graphs via Feature Alignment (github.com)](https://github.com/THUDM/GraphAlign)

[3] [All in One and One for All: A Simple yet Effective Method towards Cross-domain Graph Pretraining](https://github.com/cshhzhao/GCOPE)

[4] [AnyGraph](https://github.com/HKUDS/AnyGraph)

**Questions:**

See the weaknesses.

---

### Meta-Review · Area_Chair_nUD7 · 2024-12-10

**Metareview:**

This paper presents a new position encoding strategy for graph machine learning, although the authors claim that they focus on training the graph foundation model. This position encoding is obtained by training a model, which predicts the structural property, which I believe is not novel. This paper possesses many serious issues. Firstly, the novelty is very limited, and the contribution is overclaimed. Secondly, the correctness of the proposed method is not comprehensively justified by experiments. Only PFA is employed as baselines. Thirdly, the motivation, especially the example, seems confusing. Thus, the current version is not above the acceptance threshold.

**Additional Comments On Reviewer Discussion:**

The authors do not provide any feedback. Thus, reviewers tend to keep their ratings.

---

### Decision · Program_Chairs · 2025-01-22

Reject